# Evaluating the statistical significance of biclusters

**Jason D. Lee, Yuekai Sun, and Jonathan Taylor**
Institute of Computational and Mathematical Engineering
Stanford University
Stanford, CA 94305
{jdl17,yuekai,jonathan.taylor}@stanford.edu

## Abstract

Biclustering (also known as submatrix localization) is a problem of high practical relevance in exploratory analysis of high-dimensional data. We develop a framework for performing statistical inference on biclusters found by score-based algorithms. Since the bicluster was selected in a data dependent manner by a biclustering or localization algorithm, this is a form of *selective inference*. Our framework gives exact (non-asymptotic) confidence intervals and p-values for the significance of the selected biclusters.

## 1 Introduction

Given a matrix $X \in \mathbf{R}^{m \times n}$, *biclustering* or *submatrix localization* is the problem of identifying a subset of the rows and columns of $X$ such that the bicluster or submatrix consisting of the selected rows and columns are "significant" compared to the rest of $X$. An important application of biclustering is the identification of significant genotype-phenotype associations in the (unsupervised) analysis of gene expression data. The data is usually represented by an expression matrix $X$ whose rows correspond to genes and columns correspond to samples. Thus genotype-phenotype associations correspond to salient submatrices of $X$. The location and significance of such biclusters, in conjunction with relevant clinical information, give preliminary results on the genetic underpinnings of the phenotypes being studied.

More generally, given a matrix $X \in \mathbf{R}^{m \times n}$ whose rows correspond to variables and columns correspond to samples, biclustering seeks sample-variable associations in the form of salient submatrices. Without loss of generality, we consider square matrices $X \in \mathbf{R}^{n \times n}$ of the form

$$
\begin{aligned}
X &= M + Z, \quad Z_{ij} \sim \mathcal{N}(0, \sigma^2) \\
M &= \mu e_{I_0} e_{J_0}^T, \quad \mu \geq 0, I_0, J_0 \subset [n].
\end{aligned}
\tag{1.1}
$$

The components of $e_I, I \subset [n]$ are given by

$$
(e_I)_i = \begin{cases} 1 & i \in I \\ 0 & \text{otherwise} \end{cases}.
$$

For our theoretical results, we assume the size of the embedded submatrix $|I_0| = |J_0| = k$ and the noise variance $\sigma^2$ is known.

The biclustering problem, due to its practical relevance, has attracted considerable attention. Most previous work focuses on finding significant submatrices. A large class of algorithms for biclustering are score-based, i.e. they search for submatrices that maximize some score function that measures the "significance" of a submatrix. In this paper, we focus on evaluating the significance of submatrices found by score-based algorithms for biclustering. More precisely, let $I(X), J(X) \subset [n]$ be a (random) pair output by a biclustering algorithm. We seek to test whether the localized submatrix

$X_{I(X),J(X)}$ contains any signal, i.e. test the hypothesis

$$H_0 : \sum_{\substack{i \in I(X) \\ j \in J(X)}} M_{ij} = 0. \tag{1.2}$$

Since the hypothesis depends on the (random) output of the biclustering algorithm, this is a form of *selective inference*. The distribution of the test statistic $\sum_{\substack{i \in I(X) \\ j \in J(X)}} X_{ij}$ depends on the specific algorithm, and is extremely difficult to derive for many heuristic biclustering algorithms.

Our main contribution is to test whether a biclustering algorithm has found a statistically significant bicluster. The tests and confidence intervals we construct are exact, meaning that in finite samples the type 1 error is exactly $\alpha$.

This paper is organized as follows. First, we review recent work on biclustering and related problems. Then, in section 2, we describe our framework for performing inference in the context of a simple biclustering algorithm based on a scan statistic. We show

1. the framework gives exact (non-asymptotic) $\mathrm{Unif}(0,1)$ p-values under $H_0$, and the p-values can be "inverted" to form confidence intervals for the amount of signal in $X_{I(X),J(X)}$.

2. under the minimax signal-to-noise ratio (SNR) regime $\mu \gtrsim \sqrt{\frac{\log n}{k}}$, the test has full asymptotic power .

In section 4, we show the framework handles more computationally tractable biclustering algorithms, including a greedy algorithm originally proposed by Shabalin et al. [12]. In the supplementary materials, we discuss the problem in the more general setting where there are multiple emnbedded submatrices. Finally, we present experimental validation of the various tests and biclustering algorithms.

## 1.1 Related work

A slightly easier problem is submatrix detection: test whether a matrix has an embedded submatrix with nonzero mean [1, 4]. This problem was recently studied by Ma and Wu [11] who characerized the minimum signal strength $\mu$ for any test and any *computationally tractable* test to reliably detect an embedded submatrix.

We emphasize that the problem we consider is not the submatrix detection problem, but a complementary problem. Submatrix detection asks whether there are any hidden row-column associations in a matrix. We ask whether a submatrix selected by a biclustering algorithm captures the hidden association(s). In practice, given a matrix, a practitioner might perform (in order)

1. submatrix detection: check for a hidden submatrix with elevated mean.
2. submatrix localization: attempt to find the hidden submatrix.
3. selective inference: check whether the selected submatrix captures any signal.

We focus on the third step in the pipeline. Results on evaluating the significance of selected submatrices are scarce. The only result we know of is by Bhamidi, Dey and Nobel, who characterized the asymptotic distribution of the largest $k \times k$ average submatrix in Gaussian random matrices [6]. Their result may be used to form an asymptotic test of (1.2).

The submatrix localization problem, due to its practical relevance, has attracted considerable attention [5, 2, 3]. Most prior work focuses on finding significant submatrices. Broadly speaking, submatrix localization procedures fall into one of two types: score-based search procedures and spectral algorithms. The main idea behind the score-based approach to submatrix localization is significant submatrices should maximize some score that measures the "significance" of a submatrix, e.g. the average of its entries [12] or the goodness-of-fit of a two-way ANOVA model [8, 9]. Since there are exponentially many submatrices, many score-based search procedure use heuristics to reduce the search space. Such heuristics are not guaranteed to succeed, but often perform well in practice. One of the purposes of our work is to test whether a heuristic algorithm has identified a significant submatrix.

The submatrix localization problem exhibits a statistical and computational trade-off that was first studied by Balakrishnan et al. [5]. They compare the SNR required by several computationally efficient algorithms to the minimax SNR. Recently, Chen and Xu [7] study the trade-off when there are several embedded submatrices. In this more general setting, they show the SNR required by convex relaxation is smaller than the SNR required by entry-wise thresholding. Thus the power of convex relaxation is in separating clusters/submatrices, not in identifying one cluster/submatrix.

## 2 A framework for evaluating the significance of a submatrix

Our main contribution is a framework for evaluating significance of a submatrix selected by a bi-clustering algorithm. The framework allows us to perform *exact* (non-asymptotic) inference on the selected submatrix. In this section, we develop the framework on a (very) simple score-based algorithm that simply outputs the largest average submatrix. At a high level, our framework consists of characterizing the *selection event* $\{(I(X), J(X)) = (I, J)\}$ and applying the key distributional result in [10] to obtain a pivotal quantity.

### 2.1 The significance of the largest average submatrix

To begin, we consider performing inference on output of the simple algorithm that simply returns the $k \times k$ submatrix with largest sum. Let $\mathscr{S}$ be the set of indices of all $k \times k$ submatrices of $X$, i.e. $\mathscr{S} = \{(I, J) \mid I, J \subset [n], |I| = |J| = k\}$. The Largest Average Submatrix (LAS) algorithm returns a pair $(I_{\mathrm{LAS}}(X), J_{\mathrm{LAS}}(X))$

$$(I_{\mathrm{LAS}}(X), J_{\mathrm{LAS}}(X)) = \arg\max_{(I,J) \in \mathscr{S}} e_I^T X e_J$$

The optimal value $S_{(1)} = \mathrm{tr}\left(e_{J_{\mathrm{LAS}}(X)} e_{I_{\mathrm{LAS}}(X)}^T X\right)$ is distributed like the maxima of $\binom{n}{k}^2$ (correlated) normal random variables. Although results on the asymptotic distribution ($k$ fixed, $n$ growing) of $S_{(1)}$ (under $H_0 : \mu = 0$) are known (e.g. Theorem 2.1 in [6]), we are not aware of any results that characterizes the finite sample distribution of the optimal value. To avoid this pickle, we condition on the selection event

$$E_{\mathrm{LAS}}(I, J) = \{(I_{\mathrm{LAS}}(X), J_{\mathrm{LAS}}(X)) = (I, J)\} \tag{2.1}$$

and work with the distribution of $X \mid \{(I_{\mathrm{LAS}}(X), J_{\mathrm{LAS}}(X)) = (I, J)\}$.

We begin by making a key observation. The selection event given by (2.1) is equivalent to $X$ satisfying a set of linear inequalities given by

$$\mathrm{tr}\left(e_J e_I^T X\right) \geq \mathrm{tr}\left(e_{J'} e_{I'}^T X\right) \text{ for any } (I', J') \in \mathscr{S} \setminus (I, J). \tag{2.2}$$

Thus the selection event is equivalent to $X$ falling in the polyhedral set

$$C_{\mathrm{LAS}}(I, J) = \left\{X \in \mathbf{R}^{n \times n} \mid \mathrm{tr}\left(e_J e_I^T X\right) \geq \mathrm{tr}\left(e_{J'} e_{I'}^T X\right) \text{ for any } (I', J') \in \mathscr{S} \setminus (I, J)\right\}. \tag{2.3}$$

Thus, $X \mid \{(I_{\mathrm{LAS}}(X), J_{\mathrm{LAS}}(X)) = (I, J)\} = X \mid \{X \in C_{\mathrm{LAS}}(I, J)\}$ is a *constrained Gaussian* random variable.

Recall our goal was to perform inference on the amount of signal in the selected submatrix $X_{I_{\mathrm{LAS}}(X), J_{\mathrm{LAS}}(X)}$. This task is akin to performing inference on the mean parameter[1] of a constrained Gaussian random variable, namely $X \mid \{X \in C_{\mathrm{LAS}}(I, J)\}$. We apply the selective inference framework by Lee et al. [10] to accomplish the task.

Before we delve into the details of how we perform inference on the mean parameter of a constrained Gaussian random variable, we review the key distribution result in [10] concerning constrained Gaussian random variables.

**Theorem 2.1.** *Consider a Gaussian random variable $y \in \mathbf{R}^n$ with mean $\nu \in \mathbf{R}^n$ and covariance $\Sigma \in \mathbf{S}_{++}^{n \times n}$ constrained to a polyhedral set*

$$C = \{x \in \mathbf{R}^p \mid Ay \leq b\} \text{ for some } A \in \mathbf{R}^{m \times n}, b \in \mathbf{R}^m.$$

*Let $\eta \in \mathbf{R}^n$ represent a linear function of y. Define $\alpha = \frac{A\Sigma\eta}{\eta^T\Sigma\eta}$ and*

$$\mathcal{V}^+(y) = \sup_{j:\alpha_j<0} \frac{1}{\alpha_j}(b_j - (Ay)_j + \alpha_j\eta^Ty) \tag{2.4}$$

$$\mathcal{V}^-(y) = \inf_{j:\alpha_j>0} \frac{1}{\alpha_j}(b_j - (Ay)_j + \alpha_j\eta^Ty) \tag{2.5}$$

$$V^0(y) = \inf_{j:\alpha_j=0} b_j - (Ay)_j \tag{2.6}$$

$$F(x,\nu,\sigma^2,a,b) = \frac{\Phi\left(\frac{x-\nu}{\sigma}\right) - \Phi\left(\frac{a-\nu}{\sigma}\right)}{\Phi\left(\frac{b-\nu}{\sigma}\right) - \Phi\left(\frac{a-\nu}{\sigma}\right)}. \tag{2.7}$$

*The expression $F(\eta^Ty, \eta^T\nu, \eta^T\Sigma\eta, \mathcal{V}^-(y), \mathcal{V}^+(y))$ is a pivotal quantity with a $\mathrm{Unif}(0,1)$ distribution, i.e.*

$$F\left(\eta^Ty, \eta^T\nu, \eta^T\Sigma\eta, \mathcal{V}^-(y), \mathcal{V}^+(y)\right) \mid \{Ay \le b\} \sim \mathrm{Unif}(0,1). \tag{2.8}$$

**Remark 2.2.** *The truncation limits $\mathcal{V}^+(y)$ and $\mathcal{V}^-(y)$ (and $V^0(y)$) depend on $\eta$ and the polyhedral set $C$. We omit the dependence to keep our notation manageable.*

Recall $X \mid \{E_{\mathrm{LAS}}(I,J)\}$ is a constrained Gaussian random variable (constrained to the polyhedral set $C_{\mathrm{LAS}}(I,J)$ given by (2.3)). By Theorem 2.1 and the characterization of the selection event $E_{\mathrm{LAS}}(I,J)$, the random variable

$$F\left(S_{(1)}, \mathrm{tr}\left(e_Je_I^TM\right), \sigma^2k^2, \mathcal{V}^-(X), \mathcal{V}^+(X)\right) \mid \{E_{\mathrm{LAS}}(I,J)\},$$

where $\mathcal{V}^+(X)$ and $\mathcal{V}^-(X)$ (and $V^0(X)$) are evaluated on the polyhedral set $C_{\mathrm{LAS}}(I,J)$, is uniformly distributed on the unit interval. The mean parameter $\mathrm{tr}\left(e_Je_I^TM\right)$ is the amount of signal captured by $X_{I,J}$:

$$\mathrm{tr}\left(e_Je_I^TM\right) = |I \cap I_0|\,|J \cap J_0|\,\mu.$$

What are $\mathcal{V}^+(X)$ and $\mathcal{V}^-(X)$? Let $E_{I',J'} = e_I'e_J'^T$ for any $I',J' \subset [n]$. For convenience, we index the constraints (2.2) by the pairs $(I',J')$. The term $\alpha_{I',J'}$ is given by $\alpha_{I',J'} = \frac{|I\cap I'|\,|J\cap J'|-k^2}{k^2}$. Since $|I \cap I'|\,|J \cap J'| < k^2$, $\alpha_{I',J'}$ is negative for any $(I',J') \in \mathscr{S}_{n,k} \setminus (I,J)$, and the upper truncation limit $\mathcal{V}^+(X)$ is $\infty$. The lower truncation limit $\mathcal{V}^-(X)$ simplifies to

$$\mathcal{V}^-(X) = \max_{(I',J'):\alpha_{I',J'}<0} \mathrm{tr}\left(E_{I,J}^TX\right) - \frac{k^2\,\mathrm{tr}\left((E_{I,J}-E_{I',J'})^TX\right)}{k^2 - |I\cap I'|\,|J\cap J'|}. \tag{2.9}$$

We summarize the developments thus far in a corollary.

**Corollary 2.3.** *We have*

$$F\left(S_{(1)}, \mathrm{tr}\left(e_Je_I^TM\right), k^2\sigma^2, \mathcal{V}^-(X), \infty\right) \mid \{E_{\mathrm{LAS}}(I,J)\} \sim \mathrm{Unif}(0,1) \tag{2.10}$$

$$\mathcal{V}^-(X) = \max_{(I',J'):\alpha_{I',J'}<0} \mathrm{tr}\left(E_{I,J}^TX\right) - \frac{k^2\,\mathrm{tr}\left((E_{I,J}-E_{I',J'})^TX\right)}{k^2 - |I\cap I'|\,|J\cap J'|} \tag{2.11}$$

Under the hypothesis

$$H_0 : \mathrm{tr}\left(e_{J_{\mathrm{LAS}}(X)}e_{I_{\mathrm{LAS}}(X)}^TM\right) = 0, \tag{2.12}$$

we expect

$$F\left(S_{(1)}, 0, k^2\sigma^2, \mathcal{V}^-(X), \infty\right) \mid \{E_{\mathrm{LAS}}(I,J)\} \sim \mathrm{Unif}(0,1)$$

Thus $1 - F\left(S_{(1)}, 0, k^2\sigma^2, \mathcal{V}^-(X), \infty\right)$ is a p-value for the hypothesis (2.12). Under the alternative, we expect the selected submatrix to be (stochastically) larger than under the null. Thus rejecting $H_0$ when the p-value is smaller than $\alpha$ is an *exact* $\alpha$ level test for $H_0$; i.e. $\mathbf{Pr}_0 \left(\text{reject } H_0 \mid \{E_{\mathrm{LAS}}(I,J)\}\right) = \alpha$. Since the test controls Type I error at $\alpha$ for all possible selection events (i.e. all possible outcomes of the LAS algorithm), the test also controls Type I error *unconditionally*:

$$\mathbf{Pr}_0\left(\text{reject } H_0\right) = \sum_{I,J\subset[n]} \mathbf{Pr}_0\left(\text{reject } H_0 \mid \{E_{\mathrm{LAS}}(I,J)\}\right) \mathbf{Pr}_0\left(\{E_{\mathrm{LAS}}(I,J)\}\right)$$

$$\le \alpha \sum_{I,J\subset[n]} \mathbf{Pr}_0\left(\{E_{\mathrm{LAS}}(I,J)\}\right) = \alpha.$$

Thus the test is an exact $\alpha$-level test of $H_0$. We summarize the result in a Theorem.

**Theorem 2.4.** *The test that rejects when*

$$F\left(S_{(1)}, 0, k^2\sigma^2, \mathcal{V}^-(X), \infty\right) \geq 1 - \alpha$$

$$\mathcal{V}^-(X), = \max_{(I',J'):\alpha_{I',J'}<0} \operatorname{tr}\left(E_{I_{\mathrm{LAS}(X)}, J_{\mathrm{LAS}(X)}}^T X\right) - \frac{k^2 \operatorname{tr}\left(\left(E_{I_{\mathrm{LAS}(X)}, J_{\mathrm{LAS}(X)}} - E_{I',J'}\right)^T X\right)}{k^2 - \left|I_{\mathrm{LAS}(X)} \cap I'\right|\left|J_{\mathrm{LAS}(X)} \cap J'\right|},$$

*is a valid $\alpha$-level test for $H_0 : \sum_{\substack{i \in I(X) \\ j \in J(X)}} M_{ij} = 0$.*

To obtain confidence intervals for the amount of signal in the selected submatrix, we "invert" the pivotal quantity given by (2.10). By Corollary 2.3, the interval

$$\left\{\nu \in \mathbf{R} : \frac{\alpha}{2} \leq F\left(S_{(1)}, \nu, k^2\sigma^2, \mathcal{V}^-(X), \infty\right) \leq 1 - \frac{\alpha}{2}\right\} \tag{2.13}$$

is an *exact* $1 - \alpha$ confidence interval for $\sum_{\substack{i \in I(X) \\ j \in J(X)}} M_{ij}$. When $(I_{\mathrm{LAS}}(X), J_{\mathrm{LAS}}(X)) = (I_0, J_0)$, (2.13) is a confidence interval for $\mu$. Like the test given by Lemma 2.4, the confidence intervals given by (2.13) are also valid unconditionally.

## 2.2 Power under minimax signal-to-noise ratio

In section 2, we derived an exact (non-asymptotically valid) test for the hypothesis (2.12). In this section, we study the power of the test. Before we delve into the details, we review some relevant results to place our result in the correct context.

Balakrishnan et al. [5] show $\mu$ must be at least $\Theta\left(\sigma\sqrt{\frac{\log(n-k)}{k}}\right)$ for any algorithm to succeed (find the embedded submatrix) with high probability. They also show the LAS algorithm is minimax rate optimal; i.e. the LAS algorithm finds the embedded submatrix with probability $1 - \frac{4}{n-k}$ when $\mu \geq 4\sigma\sqrt{\frac{2\log(n-k)}{k}}$. We show that the test given by Theorem 2.4 has asymptotic full power under the same signal strength. The proof is given in the appendix.

**Theorem 2.5.** *Let $\mu = C\sqrt{\frac{2\log(n-k)}{k}}$. When $C > \max\left(\frac{1}{\sqrt{\alpha\log(n-k)}\left(\sqrt{k}-5/4\right)}, 4 + 4\sqrt{\frac{\log\frac{2}{\alpha}}{\log(n-k)}}\right)$ and $k \leq \frac{n}{2}$, the $\alpha$-level test given by Corollary 2.3 has power at least $1 - \frac{5}{n-k}$; i.e.*

$$\mathbf{Pr}(\text{reject } H_0) \geq 1 - \frac{5}{n-k}.$$

*Further, for any sequence $(n, k)$ such that $n \to \infty$, when $C > 4$, and $k \leq \frac{n}{2}$, $\mathbf{Pr}(\text{reject } H_0) \to 1$.*

## 3  General scan statistics

Although we have elected to present our framework in the context of biclustering, the framework readily extends to *scan statistics*. Let $z \sim \mathcal{N}(\mu, \Sigma)$, where $\mathbf{E}[z]$ has the form

$$\mathbf{E}[z_i] = \begin{cases} \mu & i \in S \\ 0 & \text{otherwise} \end{cases} \quad \text{for some } \mu > 0 \text{ and } S \subset [\,n\,].$$

The set $S$ belongs to a collection $\mathcal{C} = \{S_1, \dots, S_N\}$. We decide which index set in $\mathcal{C}$ generated the data by

$$\hat{S} = \arg\max_{S \in \mathcal{C}} \sum_{i \in S} z_i. \tag{3.1}$$

Given $\hat{S}$, we are interested in testing the null hypothesis

$$H_0 : \mathbf{E}[z_{\hat{S}}] = 0. \tag{3.2}$$

To perform exact inference for the selected effect $\mu_{\hat{S}}$, we must first characterize the selection event. We observe that the selection event $\{\hat{S} = S\}$ is equivalent to $X$ satisfying a set of linear inequalities given by

$$e_S^T z \geq e_{S'}^T z \text{ for any } S' \in \mathcal{C} \setminus S. \tag{3.3}$$

Given the form of the constraints (3.3),

$$a_{S'} = \frac{(e_{S'} - e_S)^T e_S}{e_S^T e_S} = \frac{1}{|S|} \left( |S \cap S'| - |S| \right) \text{ for any } S' \in \mathcal{C} \setminus S.$$

Since $|S \cap S'| \leq |S|$, we have $a_{S'} \in [-1, 0]$, which implies $\mathcal{V}^+(z) = \infty$. The term $\mathcal{V}^-(z)$ also simplifies:

$$\mathcal{V}^-(z) = \sup_{S'} \frac{1}{a_{S'}} ((e_S - e_{S'})^T z + a_{S'} e_S^T z) = e_S^T z + \sup_{S'} \frac{1}{a_{S'}} ((e_S - e_{S'})^T z).$$

Let $y_{(1)}, y_{(2)}$ be the largest and second largest scan statistics. We have

$$\mathcal{V}^-(z) \leq z_{(1)} + \sup_{S'}((e_{S'} - e_S)^T z) = z_{(1)} + z_{(2)} - z_{(1)} = z_{(2)}.$$

Intuitively, the pivot will be large (the p-value will be small), when $e_S^T z$ exceeds the lower truncation limit $\mathcal{V}^-$ by a large margin. Since the second largest scan statistic is an upper bound for the lower truncation limit, the test will reject when $y_{(1)}$ exceeds $y_{(2)}$ by a large margin.

**Theorem 3.1.** *The test that rejects when*

$$F\left(z_{(1)}, 0, k^2\sigma^2, \mathcal{V}^-(z), \infty\right) \geq 1 - \alpha$$

*where $\mathcal{V}^-(X) = e_{\hat{S}}^T z + \sup_{S'} \frac{1}{a_{S'}}((e_{\hat{S}} - e_{S'})^T z)$, is a valid $\alpha$-level test for $H_0 : e_{\hat{S}}^T \mu = 0$.*

To our knowledge, most precedures for obtaining valid inference on scan statistics require careful characterization of the asymptotic distribution of $e_{\hat{S}}^T z$. Such results are usually only valid when the components of $z$ are independent with identical variances (e.g. see [6]), and can only be used to test the global null: $H_0 : \mathbf{E}[z] = 0$. Our framework not only relaxes the independence and homoskedastic assumption, but also allows us to for confidence intervals for the selected effect size.

## 4 Extensions to other score-based approaches

Returning to the submatrix localization problem, we note that the framework described in section 2 also readily handles other score-based approaches, as long as the scores are affine functions of the entries. The main idea is to partition $\mathbf{R}^{n \times n}$ into non-overlapping regions that corresponding to a possible outcomes of the algorithm; i.e. the event that the algorithm outputs a particular submatrix is equivalent to $X$ falling in the corresponding region of $\mathbf{R}^{n \times n}$. In this section, we show how to perform exact inference on biclusters found by more computationally tractable algorithms.

### 4.1 Greedy search

Searching over all $\binom{n}{k}^2$ submatrices to find the largest average submatrix is computationally intractable for all but the smallest matrices. Here we consider a family of heuristics based on a greedy search algorithm proposed by Shabalin et al. [12] that looks for "local" largest average submatrices. Their approach is widely used to discover genotype-phenotype associations in high-dimensional gene expression data. Here the score is simply the sum of the entries in a submatrix.

---

**Algorithm 1** Greedy search algorithm

1: **Initialize:** select $J^0 \subset [n]$.
2: **repeat**
3:     $I^{l+1} \leftarrow$ the indices of the rows with the largest column sum in $J^l$
4:     $J^{l+1} \leftarrow$ the indices of the columns with the largest row sum in $I^{l+1}$
5: **until** convergence

---

To adapt the framework laid out in section 2 to the greedy search algorithm, we must characterize the selection event. Here the selection event is the "path" of the greedy search:

$$E_{\mathrm{GrS}} = E_{\mathrm{GrS}}\left((I^1, J^1), (I^2, J^2), \dots\right)$$

is the event the greedy search selected $(I^1, J^1)$ at the first step, $(I^2, J^2)$ at the second step, etc.

In practice, to ensure stable performance of the greedy algorithm, Shabalin et al. propose to run the greedy search with random initialization 1000 times and select the largest local maximum. Suppose the $m^\star$-th greedy search outputs the largest local maximum. The selection event is

$$E_{\mathrm{GrS},1} \cap \cdots \cap E_{\mathrm{GrS},1000} \cap \left\{ m^\star = \arg\max_{m=1,\ldots,1000} e_{I_{\mathrm{GrS},m}(X)}^T X e_{J_{\mathrm{GrS},m}(X)} \right\}$$

where

$$E_{\mathrm{GrS},m} = E_{\mathrm{GrS}}\left( (I_m^1, J_m^1), (I_m^2, J_m^2), \ldots \right), m = 1, \ldots, 1000$$

is the event the $m$-th greedy search selected $(I_m^1, J_m^1)$ at the first step, $(I_m^2, J_m^2)$ at the second step, etc.

An alternative to running the greedy search with random initialization many times and picking the largest local maximum is to initialize the greedy search intelligently. Let $J_{\mathrm{greedy}}(X)$ be the output of the intelligent initialization. The selection event is given by

$$E_{\mathrm{GrS}} \cap \left\{ J_{\mathrm{greedy}}(X) = J^0 \right\}, \tag{4.1}$$

where $E_{\mathrm{GrS}}$ is the event the greedy search selected $(I^1, J^1)$ at the first step, $(I^2, J^2)$ at the second step, etc. The intelligent initialization selects $J^0$ when

$$e_{[n]}^T X e_j \geq e_{[n]}^T X e_{j'} \text{ for any } j \in J^0, j' \in [n] \setminus J^0, \tag{4.2}$$

which corresponds to selecting the $k$ columns with largest sum. Thus the selection event is equivalent to $X$ falling in the polyhedral set

$$C_{\mathrm{GrS}} \cap \left\{ X \in \mathbf{R}^{n \times n} \mid \mathrm{tr}\left( e_j e_{[n]}^T X \right) \geq \mathrm{tr}\left( e_{j'} e_{[n]}^T X \right) \text{ for any } j \in J^0, j' \in [n] \setminus J^0 \right\},$$

where $C_{\mathrm{GrS}}$ is the constraint set corresponding to the selection event $E_{\mathrm{GrS}}$ (see Appendix for an explicit characterization).

## 4.2 Largest row/column sum test

An alternative to running the greedy search is to use a test statistic based off choosing the $k$ rows and columns with largest sum. The largest row/column sum test selects a subset of columns $J^0$ when

$$e_{[n]}^T X e_j \geq e_{[n]}^T X e_{j'} \text{ for any } j \in J^0, j' \in [n] \setminus J^0 \tag{4.3}$$

which corresponds to selecting the $k$ columns with largest sum. Similarly, it selects rows $I^0$ with largest sum. Thus the selection event for initialization at $(I^0, J^0)$ is equivalent to $X$ falling in the polyhedral set

$$\left\{ X \in \mathbf{R}^{n \times n} \mid \mathrm{tr}\left( e_j e_{[n]}^T X \right) \geq \mathrm{tr}\left( e_{j'} e_{[n]}^T X \right) \text{ for any } j \in J^0, j' \in [n] \setminus J^0 \right\}$$

$$\cap \left\{ X \in \mathbf{R}^{n \times n} \mid \mathrm{tr}\left( e_i e_{[n]}^T X \right) \geq \mathrm{tr}\left( e_{i'} e_{[n]}^T X \right) \text{ for any } i \in I^0, i' \in [n] \setminus I^0 \right\}. \tag{4.4}$$

The procedure of selecting the $k$ largest rows/columns was analyzed in [5]. They proved that when $\mu \geq 4/k \sqrt{n \log(n-k)}$ the procedure recovers the planted submatrix. We show a similar result for the test statistic based off the intelligent initialization

$$F\left( \mathrm{tr}\left( e_{J^0(X)} e_{I^0(X)}^T X \right), 0, \sigma^2 k^2, V^-(X), V^+(X) \right). \tag{4.5}$$

Under the null of $\mu = 0$, the statistic (4.5) is uniformly distributed, so type 1 error is controlled at level $\alpha$. The theorem below shows that this computationally tractable test has power tending to 1 for $\mu > \frac{4}{k}\sqrt{n \log(n-k)}$.

**Theorem 4.1.** *Let* $\mu = \frac{C}{k}\sqrt{n \log(n-k)}$. *Assume that* $n \geq 2\exp(1)$ *and* $n \geq \frac{k}{2}$. *When* $C > \max\left( 4\sqrt{1 + \frac{1}{4n^2}} + \frac{2}{n}, \frac{2\log 2/\alpha}{\sqrt{\log(n-k)}} + \frac{\sqrt{2}}{n} \right)$, *the* $\alpha$-level test given by Corollary 2.3 has power at least $1 - \frac{9}{n-k}$; i.e.*

$$\mathbf{Pr}(\text{reject } H_0) \geq 1 - \frac{9}{n-k}.$$

*Further, for any sequence* $(n, k)$ *such that* $n \to \infty$, *when* $C > 4$, *and* $k \leq \frac{n}{2}$, $\mathbf{Pr}(\text{reject } H_0) \to 1$.

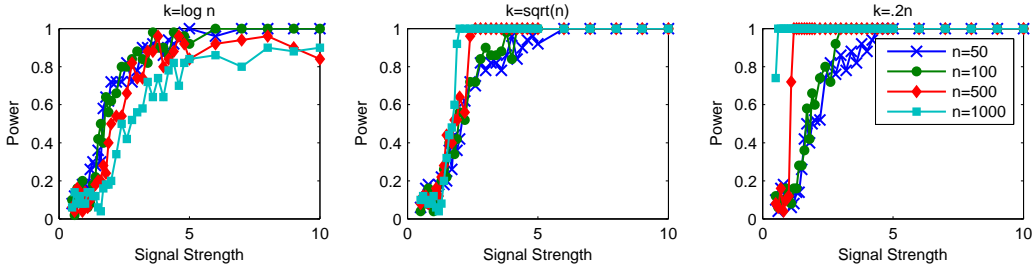

Figure 1: Random initialization with 10 restarts

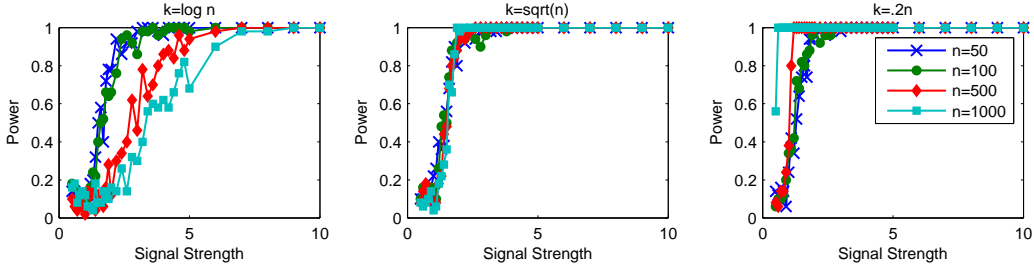

Figure 2: Intelligent initialization

In practice, we have found that initializing the greedy algorithm with the rows and columns identified by the largest row/column sum test stabilizes the performance of the greedy algorithm and preserves power. By intersecting the selection events from the largest row/column sum test and the greedy algorithm, the test also controls type 1 error. Let $\left(I_{\text{loc}(X)}, J_{\text{loc}(X)}\right)$ be the pair of indices returned by the greedy algorithm initialized with $(I^0, J^0)$ from the largest row/column sum test. The test statistic is given by

$$F\left(\text{tr}\left(e_{J_{\text{loc}(X)}} e_{I_{\text{loc}(X)}}^T X\right), 0, \sigma^2 k^2, V^-(X), V^+(X)\right), \tag{4.6}$$

where $V^+(X), V^-(X)$ are now computed using the intersection of the greedy and the largest row/column sum selection events. This statistic is also uniformly distributed under the null.

We test the performance of three of the biclustering algorithms: Algorithm 1 with the intelligent initialization in (4.4) and Algorithm 1 with 10 random restarts. We generate data from the model (1.1) for various values of $n$ and $k$. We only test the power of each procedure, since all of the algorithms discussed provably control type 1 error.

The results are in Figures 1, and 2. The $y$-axis shows power (the probability of rejecting) and the $x$-axis is rescaled signal strength $\mu \big/ \sqrt{\frac{2\log(n-k)}{k}}$. The tests were calibrated to control type 1 error at $\alpha = .1$, so any power over .1 is nontrivial. From the $k = \log n$ plot, we see that the intelligently initialized greedy procedure outperforms the greedy algorithm with a single random initialization and the greedy algorithm with 10 random initializations.

## 5 Conclusion

In this paper, we considered the problem of evaluating the statistical significance of the output of several biclustering algorithms. By considering the problem as a selective inference problem, we are able to devise exact significance tests and confidence intervals for the selected bicluster. We also show how the framework generalizes to the more practical problem of evaluating the significance of multiple biclusters. In this setting, our approach gives sequential tests that control family-wise error rate in the strong sense.

## Footnotes

[1]The mean parameter is the mean of the Gaussian prior to truncation.

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
