[Reviews · NeurIPS 2015]

Submitted by Assigned_Reviewer_1

The paper provides exact p-value for evaluating the significance of some of the score-based bi-clustering algorithms. The main result is obtained by noting that, conditioned on the selection event, data matrix $X$ is a constrained Gaussian random variable and straightforward application of the existing theorem from [10] which limits its novelty. Nevertheless, the paper provides the detailed results for score-based bi-clustering algorithms which is very relevant and comprehensive.

The paper also characterizes the asymptotic power of the studied algorithms and extends the results to more general setting of multiple embedded matrices.

Here are some minor suggestions/corrections:-

-In the related work (page 2, line 101), it would be interesting to provide brief discussion regarding the spectral-based algorithms as well.

- Theorem 2.1 from [10] needs to be rewritten. The polyhedral set is wrongly defined. Definition of $\alpha$ seems erroneous (See pages 9-10 of 10). Also please define the standard CDF $\Phi$. Please provide context to $a$ and $b$ which are the truncating parameters. Also $I_0$ and $J-0$ are not introduced in line 186 of p. 4.

- On the simulation result section on page 8 lines 412 and 420, the results are provided for three settings but have been presented for two algorithms only. It is also not clear how the intelligent initialization is outperforming the random initialization for n= 500 or n = 1000 in the k= log n case and better presentation of the graphs may help.
Summary: Interesting paper; While the novelty is limited, the analysis is quite interesting.

Submitted by Assigned_Reviewer_2

The authors provide a framework for constructing p-values for the output of a variety of "selection" tasks, focusing on biclusters in particular. Their approach is to characterize the distribution of test statistics as multivariate normal random variables constrained to linear polytopes. This framework allows them to construct exact tests (assuming normality) for subsets of variables that have been selected in different ways.

Quality: This is a quality paper. The contribution is non-trivial, and the material is presented in a way that goes beyond solving "just the bicluster significance problem." (Which is important in and of itself.) Clarity: Good but I have suggestions for improvement; see below. Originality: I believe that the specialization of theorem from reference [10] to biclustering and scan statistics and the rates that go with are novel. Significance: Addressing statistical significance issues in this domain is very important and widely applicable.

-Specific comments:

028: '"significant" compared to the rest of X. 063: "...statistically significant bicluster." 088: "We ask whether a submatrix selected by a biclustering algorithm captures the hidden association(s)." I think captures "some" of the hidden association(s) would be more accurate.

The inference is that there is *something* in the computed bicluster that has some signal. In particular, supposing that rows correspond to genes, the specified alpha level does not control false discovery of genes within the computed bicluster; it controls false discovery of an "empty" bicluster. (Similarly, "power" does not indicate the probability of having found the entire bicluster, but only a part of it.) I think it is extremely important to make this clear. If on the other hand I am mistaken about this, I think other readers would be as well.

039: Z ~ N(...) I have to ask the cliched question, what about non-normal noise models? How sensitive is the procedure to different kinds of non-normality? Depends on k maybe?

071: "...for the amount of the signal in..." I suggest explicitly defining the selected parameter, maybe before (1.2), so that you can state this formally. I also think it would be worth re-iterating that the procedure will trap the *selected* parameter with probability (1-\alpha), where the procedure is to run the bi-clustering algorithm, select the parameter of interest, and then compute/invert the p-value.

077: "In the supplementary materials, we discuss the problem in the more general setting where there are multiple emnbedded [SIC -- fix this] submatrices..." I recommend finding a way to lift this discussion into the main paper, even if you can't find room to give the full details. Not saying that needs to go here exactly, but somewhere in the main paper.

129: Use either the e^TXe form or the trace form, but choose one and use it consistently.

150: "The amount of signal in..." See comment above.

162: "Let eta ... a linear function of y" The "linear function" comment confused me; just let it be a vector.

168: I assume V0 is relevant in other contexts; if it's not relevant for this work perhaps omit?

162--174: There is a lot of notation overloading here that is made worse by the fact that vectors are typeset indistinguishably from scalars. In particular, see if there is a way to remove overloading in arguments of F (2.7).

378--397: These plots are difficult to distinguish, and I'm not sure what the intended message is. If it is to compare power of random initialization against intelligent initialization, then they should appear on the same plot. Then have different plots for (e.g.) 3 different n.
Summary: A novel, exact hypothesis testing framework for biclusters. Makes a solid contribution.

Author Feedback
Author rebuttal: First, we wish to thank the reviewers for their comments and suggestions. The two main concerns of the reviewers seem to be the novelty and significance of the paper.

1) Although the proposed pivot is the special case of a more general pivot, most known applications of the established result are to inference after model selection in regression. By casting the problem of inference with scan statistics as a selective inference problem, we are able to form valid p-values for any Gaussian scan statistic. Previously, the problem has been only approached asymptotically (even in the Gaussian case). The standard theorems in previous work state: " For a given scan statistic and conditions on the signal strength, then P(type 1 error ) + P( type 2 error) -> 0". These results are generally too coarse (since they combine type 1 and type 2 errors and rely on concentration results, instead of distributional results) to understand the distribution of the scan statistic, and cannot be used to compute a p-value. Furthermore these results do not apply to ad-hoc scan statistics that are frequently used, such as the greedy algorithm. We believe that providing tests with provable type 1 error control for ad-hoc procedures is a major contribution of this work. In other words, this work is concerned with testing the output of a submatrix/biclustering algorithm. As is standard in the hypothesis testing setup, we always ensure that the test controls type 1 error, and under certain assumptions on the signal strength we show that the type 2 error tends to 0.

2) To our knowledge, there are no results in the literature on the power of tests based on the general framework of Lee et al. (2013). We provide two different power results. The first is for the combinatorial procedure of Theorem 2.5. This result shows that the selective test results in no loss of power, since the signal conditions match those in Balakrishnan et al. The second power theorem applies to the largest row sum/column sum test in Theorem 4.1. This test is computationally tractable and attains the optimal signal conditions among computationally tractable procedures, assuming the planted clique conjecture ( Ma and Wu 2013). This can be seen by letting \mu=constant, which corresponds to k=sqrt(n logn). Theorem 4.1 shows that the power tends to 1 in this regime. This is optimal up to the sqrt( logn) since assuming planted clique there is no computationally tractable procedure for detecting in the regime mu=constant and k=o(sqrt(n)).

Reviewer 1:
1. Although we don't study the power of the greedy algorithm, we study the power of a computationally feasible algorithm in Theorem 4.1.

2. Yes, there is a typo in the definition of C.

Reviewer 2:
1. We will add discussion of spectral algorithms.
2. Yes, Theorem 2.1 has a typo. We will clarify $\Phi$, I_0, and J_0.
3. The reviewer is correct that when k=logn, the intelligent initialization is not helping. As we can see from Theorem 4.1, the row/column test requires that k~ sqrt(n) when mu is a constant. Thus in the3 plots of k=sqrt(n) and k=.2n , we see that the intelligent initialization is helping.

Reviewer 3:
1. We will clarify that a false discovery is an empty bicluster, and does not refer to the proportion of genes within the computed bicluster.
2. Under certain assumptions, the approach also gives asymptotically valid inference when the noise is non-Gaussian. Theoretical justification for the appeal to asymptotics is described in Tian and Taylor (2015) .
3. We will incorporate your suggestions to make the writing more clear.

X. Tian and J.E. Taylor, Asymptotics of selective inference, arXiv preprint arXiv:1501.03588.

Reviewer 5.
We will try to redo the exposition in Section 2 to help give context and motivation.

Reviewer 6.
Thank you for your comments. We agree that the importance of this work is to complement existing work that only tests for biclusters without finding them, or only provides sufficient conditions for correct estimation. Our work allows for testing what an algorithm finds.